# Decades of Genetic Research on *Soybean mosaic virus* Resistance in Soybean

**DOI:** 10.3390/v14061122

**Published:** 2022-05-24

**Authors:** Mariola Usovsky, Pengyin Chen, Dexiao Li, Aiming Wang, Ainong Shi, Cuiming Zheng, Ehsan Shakiba, Dongho Lee, Caio Canella Vieira, Yi Chen Lee, Chengjun Wu, Innan Cervantez, Dekun Dong

**Affiliations:** 1Division of Plant Science and Technology, University of Missouri, Columbia, MO 65201, USA; klepadlom@missouri.edu; 2Delta Center, Division of Plant Science and Technology, University of Missouri, Portageville, MO 63873, USA; leedongho@missouri.edu (D.L.); canellavieirac@mail.missouri.edu (C.C.V.); ylkg2@missouri.edu (Y.C.L.); 3College of Agronomy, Northwest University of Agriculture, Jiangling, Xianyang 712100, China; lidexiao@nwafu.edu.cn; 4London Research and Development Centre, Agriculture and Agri-Food Canada, London, ON N5V 4T3, Canada; aiming.wang@canada.ca; 5Department of Horticulture, University of Arkansas, Fayetteville, AR 72701, USA; ashi@uark.edu; 6Global R&D, Driscoll’s, Watsonville, CA 95076, USA; cuiming@gmail.com; 7Rice Research and Extension Center, Department of Crop, Soil, and Environmental Sciences, University of Arkansas, Stuttgart, AR 72160, USA; eshakiba@uark.edu; 8Department of Crop, Soil, and Environmental Sciences, University of Arkansas, Fayetteville, AR 72701, USA; cwu@uark.edu; 9Bayer CropScience, Global Soybean Breeding, 1781 Gavin Road, Marion, AR 72364, USA; innan.cervantes-martinez@basf.com; 10Soybean Research Institute, Zhejiang Academy of Agricultural Sciences, Hangzhou 310021, China; dekun.dong@foxmail.co

**Keywords:** *Soybean mosaic virus*, virus resistance, soybean breeding, genetic diversity

## Abstract

This review summarizes the history and current state of the known genetic basis for soybean resistance to *Soybean mosaic virus* (SMV), and examines how the integration of molecular markers has been utilized in breeding for crop improvement. SVM causes yield loss and seed quality reduction in soybean based on the SMV strain and the host genotype. Understanding the molecular underpinnings of SMV–soybean interactions and the genes conferring resistance to SMV has been a focus of intense research interest for decades. Soybean reactions are classified into three main responses: resistant, necrotic, or susceptible. Significant progress has been achieved that has greatly increased the understanding of soybean germplasm diversity, differential reactions to SMV strains, genotype–strain interactions, genes/alleles conferring specific reactions, and interactions among resistance genes and alleles. Many studies that aimed to uncover the physical position of resistance genes have been published in recent decades, collectively proposing different candidate genes. The studies on SMV resistance loci revealed that the resistance genes are mainly distributed on three chromosomes. Resistance has been pyramided in various combinations for durable resistance to SMV strains. The causative genes are still elusive despite early successes in identifying resistance alleles in soybean; however, a gene at the *Rsv4* locus has been well validated.

The history of soybean (*Glycine max*) symptoms caused by *Soybean mosaic virus* (SMV) starts in the U.S. as early as 1915, when it was first observed and documented in Western scientific literature by Clinton (1916) [1]. Several years later, Gardner and Kendrick (1921) [2] described the symptoms in detail as mosaic, dark green leaf areas with misshapen and stunted leaflets, and the discoloration of the seed coat (mottling) caused by the bleeding of hila pigment into other regions. Other symptoms included local and/or systemic necrosis [3]. Since then, many scientific studies describing the principles of SMV infection and plant resistance mechanisms have been released. Among over 100 viruses that can infect soybean, SMV is a major threat for the soybean industry [4], causing serious yield losses and the deterioration of seed quality via the reduction in seedling viability and vigor, seed coat mottling, flower abortion, and reduction in pod set, seed number, and seed size [5,6,7,8,9]. Moreover, SMV-infected seeds can lead to an increased protein and decreased oil content [10,11]. Depending on the soybean genotype and SMV strain, yield can be reduced by up to 90% in severe outbreaks [8,11,12]. The distribution of SMV disease has reached all soybean-growing countries, including Brazil, Canada, China, Japan, South Korea, and the United States [13,14,15].

*Soybean mosaic virus* is a plant pathogen of the *Potyvirus* genus within the *Potyviridae* family. The *Potyviridae* family represents approximately 30% of known plant viruses and encompasses the most economically important plant viruses infecting major crops worldwide [4,5,16,17]. The virus particle consists of a linear, positive sense, single-stranded RNA of approximately 9.6 kb, and has a genome-linked viral protein (VPg) covalently bound to the 5′ end and 3′ poly-A tail [4,18]. The RNA molecule is encapsulated in filamentous virions composed of stacked coat proteins, creating a flexuous and rod-shaped SMV particle with helical symmetry and a size of 650–700 nm in length and 15–18 nm in width [19]. The viral genome encodes a large open reading frame (ORF) translated into a large polyprotein and produces 10 different functional proteins derived from a proteolytic reaction. In addition, with a frameshift in the P3 cistron, the genome also produces a small ORF encoding another functional protein (11th protein) [20,21]. These 11 mono- or multi-functional proteins are P1, HC-Pro, P3, P3N-PIPO, 6K1, CI, 6K2, NIa-VPg, NIa-Pro, NIb, and CP [18,20,22,23,24,25].

SMV can be transmitted to soybean plants through three methods. The virus can translocate through infected seed from region to region and generation to generation in approximately 50 plant species in the *Leguminosae*, *Fabaceae*, *Amaranthaceae*, *Chenopodiaceae*, *Passifloraceae*, *Schropulariaceae*, and *Solanaceae* families [26,27]. Secondly, the virus can be transmitted by over 30 aphid species in a non-persistent manner; aphids must acquire the virus by regularly feeding on infected plants within a season [26,28,29]. Lastly, tissue damage could potentially be an entry site for SMV infection. Transmission via mechanical wounding is commonly used for genetic studies and breeding purposes. An SMV inoculum can be prepared by grinding infected leaves with a mortar and pestle in a buffer solution (0.01 M sodium phosphate, 0.01 M potassium phosphate, or 0.05 M sodium citrate). Plants are, subsequently, inoculated with SMV by rubbing abrasive dusted leaves with a pestle dipped in the inoculum [13,30,31,32,33,34]. SMV-free seeds are difficult to achieve due to the constant transmission of the disease. Eliminating sources of SMV by roguing infected plants in the field, vector control, alternate host rotation, and weed management reduces the virus incidence [8,9,35]. SMV strains can be preserved through several techniques. For short-term storage, SMV can be maintained in vivo by infecting susceptible soybean lines and in vitro via virus-infected callus cultures [36]. For long-term storage, infected leaf tissues can be frozen at −80 °C [37,38].

In recent years, considerable progress has been contributed to uncover the genetic and molecular basis of SMV infection and host resistance. This review presents the current status of the SMV pathosystem in soybeans, with a focus on *Rsv* genes for resistance to USA SMV strains.

## 1. SMV Classification Systems

Several classification systems for grouping SMV isolates have been proposed independently around the world. Conover (1948) [39] reported that several SMV strains caused the disease, and that temperature affects symptom expression. Currently, SMV is classified into strains based on its virulence and symptoms on differential host lines [13,40]. In Japan, SMV isolates were classified into five main strains (A-E) [41,42]. SMV isolates were classified into 11 groups, G1-G7, SMV-N, G5H, G7a, and G7H, in South Korea [43,44]. In China, they were classified into 22 groups (SC1–SC22) according to geographical regions and individual soybean responses [14,45,46,47,48]. Although the isolates may be different based on various classification systems, it is still important to determine their virulence and relationships between different SMV isolates to identify new resistance genes [49,50,51].

The SMV isolate classification system in the United States was established by Cho and Goodman (1979) [13] based on reactions of two susceptible soybean cultivars (‘Clark’ (Plant Introduction (PI) 548533) and ‘Rampage’ (PI 548609)), and six resistant cultivars (‘Davis’ (PI 553039), ‘York’ (PI 553038), ‘Kwanggyo’, ‘Marshall’ (PI 548693), ‘Ogden’ (PI 548477), and ‘Buffalo’). Ninety-eight SMV isolates have been classified into seven strains (G1–G7), where G1 is the least virulent, only infecting susceptible cultivars but none of the resistant cultivars, and G7 being the most virulent, infecting resistant soybean cultivars causing necrosis in Marshall, Ogden, Kwanggyo, and Buffalo, and mosaic symptoms in Davis and York. This classification system was later revised by Chen and Choi (2008) [52] where Buffalo was replaced by PI 96983, which carries a single resistance gene conferring the same reaction as Buffalo (Table 1). This strain classification system has been widely used for SMV isolate differentiation and the prediction of resistance genes and alleles in soybean germplasm.

The virus continuously adapts and evolves, producing new isolates that overcome host resistance. Several isolates that overcame resistance have been identified over the past two decades in South Korea and Brazil. SMV-G3A and G5H were initially reported by Cho et al. (1983) [43] as mutated strains from G3 and G5, respectively. The G3A mutant induced mosaic symptoms in Marshall, Ogden, and ‘Jangbaekkong’. The G5H mutant caused necrosis in ‘Suweon 97’ and a mosaic reaction in Jangbaekkong. Mosaic symptoms were later observed by Kim and Lee (1991) [73] in Marshall and ‘Paldalkong’ after infection by G5DH. In Brazil, the SMV isolate SMV-95-1 was classified as a member of SMV-G5 [74]. In South Korea, an evolved SMV strain G7H overcame genetic resistance [75], and a new resistance-breaking isolate L-RB from the G2 pathotype was identified in Canada [23]. In recent years, the genetic analysis of SMV diversity in Iran revealed a new SMV mutation in a P3 protein, overcoming *Rsv4* resistance in the soybean cultivar ‘V94-5152’ [76]. A novel resistance-breaking SMV-R strain was found to be common in the central and southern provinces of China [77]. These reports indicated that gene-mediated resistance in soybean is not durable due to high selection pressure from the constantly evolving SMV isolates generated by viral RNA polymerase, which lacks the proof-reading ability [4]. The effective control of new and aggressive strains requires significant efforts to manage and limit disease prevalence in soybeans [78]. Therefore, disease management and new sources of resistance are essential in plant breeding and crop improvement [46,52].

## 2. Molecular Mechanisms of SMV Infection

The molecular interactions between SMV and the plant host are complex, and many mechanisms are still unknown. The virus is released directly into the host cell via the mechanical damage of soybean tissue and its infection action appears to be similar to other Potyviruses [78,79]. Upon entering a susceptible plant cell, the coat protein (CP) is removed (virion encapsidation) and the genetic information is translated. The RNA genome is a direct template for translation using the cap-independent internal ribosome entry site (IRES) for initiation [78]. Two products of translation are produced as precursors to functional proteins: a long polyprotein as a result of the translation of the entire genome, and a short polyprotein P3N-PIPO produced via a ribosomal frameshift. After translation, polyproteins are subjected to proteolytic processing by three self-encoded proteases to yield mature proteins [78,79,80].

The SMV infection cycle consists of several major steps: the entry, uncoating, translation, replication, and cell-to-cell and long-distance movement. Shortly after translation, the viral genome is replicated by NIb, an RNA-dependent RNA polymerase (RdRp), in association with cytoplasmic membranes that create a specific micro-environment to protect the viral genome from silencing [78,80,81]. The copied molecules are coated (virion assembly) and some of the viral particles move into the neighbor cells through plasmodesmata (PD). The intercellular passage requires the assistance of the CI-formed, cone-shaped structures that can increase the plasmodesmata size exclusion limit (SEL). The formation of the CI conical structures that anchor to and extend through the PD is mediated by the P3N-PIPO protein [82]. In addition to CI, CP, and P3N-PIPO, several other viral proteins, including HC-Pro and VPg, may also play a role in viral transport through plasmodesmata [82,83,84]. Long-distance movement occurs when the virus spreads through the vascular system and can infect cells located far from the initial infection point (systemic infection) [80,84,85]. However, this phloem-dependent movement is poorly understood.

SMV resistance-mediated signaling pathways in soybean remain largely unknown. Babu et al. (2008) [86] used microarray technology to detect expression changes of the ‘Williams 82’ (PI 518671) SMV-susceptible (*rsv*) genome infected by the SMV-G2 strain. Many genes that were related to hormone metabolism, cell wall biogenesis, chloroplast functions, and photosynthesis were significantly downregulated at 14 days post-inoculation. The genes involved in defense were upregulated at the late stages, suggesting that the response to SMV was delayed and the plant could not combat the infection. In addition, in research conducted by Chen et al. (2016) [87], small RNAs, degradome, and transcriptome sequencing analyses were used to identify multiple differentially expressed genes and microRNAs in Williams 82 infected by three SMV isolates and PI 96983 (resistant or necrotic to SMV; *Rsv1*) by SMV-G7. The knock-down of the eukaryotic translation initiation factor 5A (eIF5A) gene diminished necrosis and enhanced viral accumulation, suggesting its essential role in the SMV-G7-induced and *Rsv1*-mediated signaling pathway. The status of the soybean–SMV interactions has recently been summarized by Liu et al. (2016) [79].

## 3. Symptom Induction by SMV

The symptoms induced by SMV depend on many factors, including the host genotype, virus strain, plant age at infection, and environment [88]. Genetic resistance to SMV seems to be the most efficient strategy to control the disease [5,19,89]. The first and most important step in producing soybeans with SMV resistance is to identify resistant germplasm and then to study the molecular mechanisms [90,91]. Individual cultivar reactions to SMV strains are classified into three main responses: resistant (R, symptomless); necrotic (N, systemic necrosis); or susceptible (S, mosaic) (Figure 1) [13,31].

Resistant soybeans have no disease symptoms and are undistinguishable from non-infected plants. A host plant is resistant if it can block viral replication and movement (cell-to-cell or long-distance) and, therefore, cannot be detected visually or by using the enzyme-linked immunosorbent assay (ELISA) test [92]. A resistant reaction restricts the virus to the infection site and prevents disease dispersion. In a resistant reaction, the pathogen is described as avirulent, and the virus–host interaction is incompatible [80].

Necrosis is a hypersensitive (HR) protective system activated in response to SMV. In general, the necrotic reaction causes necrotic spots on inoculated leaves, the yellow and brown discoloration of upper leaves and veins, stunting of the entire plant, browning of stems and petioles, defoliation, and plant death as a system of reduction in disease within the crop [56,81,93]. The signaling pathway causing the necrotic reaction remains largely unknown. Some soybean accessions develop necrotic symptoms in a short time after infection, leading to plant death in the V2 developmental stage [56]. In necrotic plants, viral replication and cell-to-cell or long-distance movement occur at a lower level. The necrotic reaction can be classified into three subgroups: local necrosis, where virus proliferation is restricted to the initially infected cell (necrotic spots on the leaves, stems, and petioles); systemic necrosis (necrotic spots on upper leaves, browning of leaf veins, petioles, and stems); and stem-tip necrosis, when the reaction is not restricted to the initial infection site (bud blight, stunting of the plants, defoliation, and plant death) [40,81]. Necrosis is associated with environmental temperature and the heterozygous state of resistance alleles; however, local and systemic necrosis may also be observed when a homozygous resistance gene interacts with specific SMV strains [31,32,34,53,93,94]. High temperatures exceeding 37 °C tend to mask the expression of mosaic symptoms in both homozygous and heterozygous plants [93]. There has been disagreement among researchers in terms of the classification of necrotic responses as either susceptible or resistant. Some researchers consider necrosis as a susceptible reaction due to significant yield losses or plant death [95,96,97,98]. However, results from many genetic studies suggest that necrotic plants should be classified as resistant when evaluating segregating populations due to the phenotypic expression of heterozygotes [32,94,99]. In addition, the necrotic reaction is always associated with the presence of resistance genes/alleles interacting with specific SMV strains. Therefore, necrotic plants should be classified with the resistant class as they contain resistance genes regardless of the symptom expression [94,99].

Susceptible soybean genotypes develop a characteristic stunted growth and mosaic leaf pattern, leading to curling leaves, twisting leaf edges downward, and seed-coat mottling [100,101]. In susceptible plants, a substantial yield reduction has been reported when SMV infection occurs at the early stages of plant development [6,35]. A host plant is considered fully susceptible if the virus can successfully complete replication and cell-to-cell and long-distance movement [102]. When the disease symptoms are present, the pathogen is described as virulent, and the virus–host reaction is compatible [80]. There are two types of susceptible phenotypes distinguished as early and late mosaics. Early mosaic is the most common response observed as soon as 5–7 days after inoculation. Late mosaic has a 2–3 weeks delay in symptom expression [26]. Mosaic symptoms are more severe at lower temperatures, whereas higher temperatures alter stem-tip necrosis to mosaic symptoms or mask the mosaic symptoms [93,103,104].

## 4. SMV Resistance Genes

Early genetic studies on host resistance to SMV were conducted in Japan [96]; however, there has been significant progress since the establishment of the SMV classification system by Cho and Goodman (1979) [13] for SMV resistance genes and their alleles [5,32,34,37,56,60,66,67,68,70,105,106]. Soybean resistance to SMV is controlled by four nuclear genes, and there is no significant cytoplasmic effect on the SMV reaction as determined by reciprocal crosses [31,53,65].

The identification of resistance genes (R-genes) in a plant genome is based on inheritance and allelism studies [37,54,55,58,64,66]. Recombinant inbred line (RIL) populations, developed by crossing a resistant genotype and a susceptible genotype, e.g., ‘Essex’ (PI 548667) or ‘Lee 68’ (PI 559369), are the most common type of a bi-parental mapping population used to determine the source, location, and number of genetic resistance genes. On the other hand, an allelism test crosses the resistant genotype in question with a set of several differential resistant genotypes [32]. Observed segregations of F_2_ and F_2:3_ lines inoculated with SMV have been used for testing goodness-of-fit to the expected genetic ratios by the Chi-square test (χ^2^). The presence of recombinant plants indicates that the resistant parent in question carries different alleles for SMV resistance; otherwise, genes in both parents are allelic at the same locus if there is no segregation in the progeny. If a genotype in question turns out to be allelic to a reported locus, it is necessary to compare the pattern of reaction symptoms of the newly identified allele with those of known alleles to determine if the SMV reaction of the new allele is unique. Good examples of such a genetic strategy were tested in ‘Tousan 140’ [66], ‘Zao 18’ [68], ‘Raiden’ (PI 360844) [55], Suweon 97 [54], and ‘J05’ [67]. The results can be confirmed through molecular studies by applying genetic markers such as simple sequence repeat (SSR) or single-nucleotide polymorphism (SNP) that are positioned within or nearby the gene of interest [67,69].

The classical genetic analysis for virus resistance is based on the gene-for-gene hypothesis established by Flor (1955) [107]. Typically, a single dominant resistance gene of the host plant specifically interacts with the product of a single dominant avirulence gene in the pathogen. A gene-for-gene relationship exists when the presence of a gene in one population is contingent on the continued presence of a gene in another population, and when the interaction between the two genes leads to a single phenotypic expression by which the presence or absence of the relevant gene in either organism may be recognized [108]. There are different perceptions on the interaction of SMV and soybean. In China, resistance to different SMV strains was presumed to be conditioned by different genes, and there was no pleiotropic effect on a resistance gene [109]; however, in the U.S., SMV resistance genes were shown to be pleiotropic. Therefore, two different reactions to separate SMV strains may be controlled by the same host gene [110].

To date, four independent loci for SMV resistance, *Rsv1*, *Rsv3*, *Rsv4*, and *Rsv5*, have been identified [53,59,63,111] (Table 1). Most resistant soybean genotypes carry a single dominant gene [34,47,106]; however, some genotypes contain two genes for resistance [60,66,67,68,70,71]. One landrace ‘8101’, collected from South Korea, was identified to carry three SMV R-genes [72]. Pyramiding resistance genes is a technique commonly used in breeding programs to control plant pathogens. However, this was not the case for SMV due to concerns about the possible occurrence of a superior SMV isolate with the ability to overcome all *R* genes. Recent molecular analyses revealed that the resistance breakdown mechanisms for different resistance genes are diverse and simultaneous mutations in multi-viral proteins are required for SMV to gain virulence to multiple *R* genes [22,112]. Therefore, the chance for the occurrence of a superior SMV isolate that can overcome all resistance genes is extremely low, and pyramiding *R* genes may be a sound option. Indeed, the pyramiding of three R-genes using marker-assisted selection (MAS) has been reported, and potential lines may be released to support breeding programs [61,64,89,113].

## 5. *Rsv1* Locus Confer Resistance to Less Virulent SMV Strains

*Rsv1* was the first SMV resistance locus identified and mapped to chromosome (chr.) 13 (linkage group (LG) F), and it is the most common in soybean germplasm [53]. The *Rsv1* locus contains at least ten alleles: *Rsv1*, *Rsv1-t*, *Rsv1-y*, *Rsv1-m*, *Rsv1-k*, *Rsv1-r*, *Rsv1-s*, *Rsv1-n*, *Rsv1-h*, and *Rsv1-c* [31,33,53,54,55,57]. Most of these alleles exhibit partial dominance and confer resistance to less virulent strains, G1 to G3, but susceptibility or necrosis to more virulent strains, G5 to G7 (Table 1). Resistance conferred by *Rsv1* is SMV strain-specific and is associated with a typical HR and gene dosage effect associated with heterozygotes, resulting in necrotic reactions in soybean [14,31,66,71]. The P3 and HC-Pro proteins have been found to be the effectors of *Rvs1*-mediated resistance [114,115,116].

The first allele, *Rsv*, was discovered by Kiihl and Hartwig (1979) [53] in PI 96983, collected from South Korea and reassigned later as *Rsv1* by Chen et al. (1991) [31]. PI 96983 was crossed to a susceptible cultivar, and an F_2_ population indicated a dominant nature of *Rsv1*. The *Rsv1* allele in PI 96983 showed greater effectiveness in suppressing virus development when evaluated by grafting [53]. Buffalo exhibited the same reaction as PI 96983 to different strains. Isogenic lines derived from ‘Williams (6)’ × Buffalo exhibited two types of reactions: resistant to G1 and G5, and resistant to G1 but susceptible to G5. Williams was reported to be susceptible to all SMV strains [117,118]. Resistance to SMV-G1 and G7 strains in Buffalo was under monogenic control at the *Rsv1* locus [119]; therefore, it was proposed that the resistance gene in Buffalo may be a different allele at the *Rsv1* locus [120,121]. In the present SMV classification system, Buffalo was replaced by PI 96983, which carries a single dominant allele at the same locus [52].

The Ogden soybean cultivar carries an allele designated as *Rsv1-t*, previously known as *rsv^t^*, and it shows resistance to SMV strains G1, G2, and G4–G6, but necrosis when inoculated with G3 and G7 [31,47,53,66,69,71]. The ratio of necrotic to resistant plants was 1:1 in the F_2_ populations from Ogden × susceptible genotype [31,53], indicating the incomplete dominant nature of *Rsv1-t*. The heterozygous plants at *Rsv1-t* permitted greater virus development than the homozygotes. Some plants with heterozygous alleles displayed necrosis and were grouped with resistant classes [53].

The soybean cultivar York carries the *Rsv1-y* allele, and it is resistant to the less virulent strains G1–G3 and susceptible or necrotic to the more virulent strains G4–G7 [13,31,33]. Resistance to SMV in York is controlled by a single dominant gene [33], which was later designated as *Rsv1-y* [31]. When York was crossed to a susceptible genotype, nearly one-fourth of the plants observed in the F_2_ populations were necrotic. Additionally, there was a low level (about 0.6–1.3%) of necrotic tissue, but no susceptible lines were detected in the crosses York × PI 96983, York × Marshall, and York × Ogden [31]. Silva et al. (2004) [122] reported the soybean cultivar FT-10 to carry an allele at the *Rsv1* locus, probably originating from Davis, and a gene symbol of *Rsv1-d* was assigned. According to its pedigree, *Rsv1-d* should be the same as *Rsv1-y*, because Davis and York showed the same symptoms to G1 to G7 and presumably carry the same resistance gene [13,111] derived from ‘Dorman’ (PI 548653) [65]. The *Rsv1-y* in York was later proved to be a separate locus, although tightly linked to *Rsv1* (2.2 cM), and assigned a new locus *Rsv5* [65].

Chen et al. (1991) [31] confirmed that the genes in Marshall and Kwanggyo are allelic to the *Rsv1* locus and designated them as *Rsv1-m* and *Rsv1-k*, respectively. Marshall contains the single dominant allele *Rsv1-m*, and confers resistance to strains G1, G4, and G5, and necrosis to the rest of the strains [31,34]. At the homozygous state, the *Rsv1-m* in Marshall confers complete resistance or necrosis to SMV strains, but *Rsv1-k* in Kwanggyo displays a mixture of symptoms of resistance and delayed necrosis to some strains. This type of necrosis is characterized by systemic lesions and stem browning, which is different from typical stem-tip necrosis (Figure 1). *Rsv1-k* confers resistance to SMV-G1 through G4, and necrosis to SMV-G5 through G7. When Kwanggyo was crossed to a susceptible genotype, there was an equal number of necrotic and resistant plants in the F_2_ population indicating the partial dominance of *Rsv1-k* [31]. Additionally, the viral strain could not be recovered from the necrotic plants because viral replication and movement were restricted by the death of cells surrounding the infection site [31,123].

The Raiden cultivar contains a single *Rsv1-r* allele and displays resistance to SMV-G1 through G4 and G7, but necrosis to SMV-G5 and G6 [55,111]. Raiden and ‘L88-8431’, a Williams back-crossed (BC_5_) isogenic line with SMV resistance derived from Raiden, were crossed to a susceptible soybean cultivar. The F_2_ populations and F_2:3_ progenies from both crosses did not show any segregation for susceptibility and proved that the resistance to SMV in Raiden and ‘L88-8431’ was controlled by a single dominant gene allelic to *Rsv1* [55].

LR1 contains the *Rsv1-s* allele and confers resistance to SMV-G1 through G4 and G7, and systemic necrosis to SMV-G5 and G6 [34,37]. The LR1 line, derived from Essex × PI 486355, was crossed with Lee 68, Ogden, and York, and results showed a single dominant gene at the *Rsv-1* locus. The *Rsv1-s* confers partial resistance, and its expression is gene-dosage dependent, where homozygosity conferred resistance and heterozygosity conferred systemic necrosis [37].

PI 507389, released as the soybean cultivar ‘Tousan 50’, contains the *Rsv1-n* allelic to *Rsv1* and does not show any resistance, but shows necrosis when infected with G1, G2, G5, and G6 [56]. The necrosis allele *Rsv1-n* is co-dominant with the allele for susceptibility, and the heterozygotes showed a mixed phenotype of necrosis and a susceptible reaction (N/S). The resistance allele became recessive to the susceptible allele as the mixed N/S reaction shifted to susceptible at a later stage in response to more virulent strains [56,124].

Suweon 97 contains the most valuable *Rsv1-h* allele conferring resistance to all SMV strains G1-G7 and G7A [54]. Inheritance and allelic studies showed that the *Rsv1-h* allele displays complete dominance at the *Rsv1* locus [54].

Recently, a single dominant allele *Rsv1-c* has been discovered in ‘Corsica’ (PI 559931) that brings different patterns of responses to SMV strains than other genotypes with known genes. The unique pattern is characterized by early resistance (ER) at the seedling stage to G2, G5, and G7 strains, but susceptibility to G1, G3, and G6 [57].

In summary, the alleles at the *Rsv1* locus are diverse, abundant, and widely distributed [34,57,111]. Among the ten alleles of the *Rsv1* locus, *Rsv1-y* and *Rsv1-k* are most frequent in diverse germplasm [106,125]. The alleles of the *Rsv1* locus exhibit partial or complete dominance and confer resistance to some, but not all, SMV strains. The alleles at the *Rsv1* locus can be grouped into several distinct classes: (A) alleles conferring resistance to most or all SMV strains, such as *Rsv1-h* in Suweon 97 and *Rsv1* in PI 96983; (B) alleles conferring both resistance and necrosis such as *Rsv1-t*, *Rsv1-m*, *Rsv1-k*, and *Rsv1-s* alleles in Ogden, Marshall, Kwanggyo, Raiden, and LR1, respectively; (C) alleles conferring resistance or a mosaic reaction as *Rsv1-y* in York and *Rsv1-c* in Corsica; and (D) alleles conferring necrosis and a mosaic reaction as *Rsv1-n* in PI 507389. Genetic studies on *Rsv1* demonstrated that alleles with necrotic symptoms to specific SMV strains in the homozygous state are dominant to alleles resistant or susceptible to the same strain, and alleles resistant in the homozygous state often exhibit necrosis when they occur in a heterozygote with a susceptible allele [32]. The necrotic allele (*Rsv1-n*) is co-dominant with the allele for susceptibility (*rsv*).

## 6. Mapping of the *Rsv1* Locus

The *Rsv1* gene was first mapped in the PI 96983 × Lee 68 cross, where PI 96983 was the resistant parent (*Rsv1*) and Lee 68 was the susceptible parent (*rsv*), using two restriction fragment length polymorphism (RFLP) markers pA186 and pK644a, and one simple sequence repeat (SSR) marker linked to *Rsv1* with distances of 1.5, 2.1, and 0.5 cM, respectively [121]. Later, Zheng et al. (2003) [126] and Li et al. (2004) [117] reported a random amplified polymorphic DNA (RAPD) marker, OPN11, linked to *Rsv1* with a distance of 1.2–1.8 cM, and its linkage was confirmed using single-nucleotide polymorphism (SNP) markers designed from an OPN11 clone [127]. In addition, four amplified fragment length polymorphism (AFLP) markers R11 (518 bp), R12 (171 bp), R13 (261 bp), and R14 (312 bp) were identified to be linked to *Rsv1* in the F_2_ population PI 96983 (*Rsv1*) × Lee 68 (*rsv*) using a bulk segregant analysis. The markers were mapped to chr. 13 (LG F) within 3.5 cM to *Rsv1* with the closest one, R11, located at 0.4 cM [128,129]. In total, twelve NBS-LRR resistance gene analogs were mapped at the *Rsv1* locus in PI 96983 [129]. In another study, Gore et al. (2002) [120] constructed a high-resolution map of the *Rsv1* region containing 38 loci with 24 markers, including 1 RAPD (OPN11), 4 SSR (HSP176, 64-A8C, Satt510, and Satt120), and 19 RFLP markers. The *Rsv1* gene was closely linked to SSR markers HSP176 (2.9 cM), Satt510 (2.4 cM), and 64-A8C (0.5 cM) as being the closest SSR markers to *Rsv1*. One RAPD marker OPN11_980/1070_ and its derived sequence-characterized amplified region (SCAR) marker SCN11_980/1070_ were also linked to *Rsv1* with the same distance of 3.03 cM [126].

Later, four allele-derived SNPs were found in N11PF fragments amplified from the SCAR primer and validated in two F_2_ populations of PI 96983 × Lee 68, and ‘Myeongjunamulkong’ × Lee 68 [127]. Two other SNP markers were developed from the L20a fragment, a Toll and Interleukin-1 Receptor (TIR)-Nucleotide-Binding Site (NBS) genomic sequence and mapped close to *Rsv1* [130]. In another study, *Rsv1* in the J05 soybean was mapped using the F_2_ population derived from a cross of J05 (R) × Essex (S). The *Rsv1* locus was flanked by Sat_154 and Satt510 with 0.5 and 2.3 cM, respectively [69].

Hayes et al. (2004) [130] reported six clones on chr. 13 (LG F) around the *Rsv1* locus associated with SMV resistance, and three of them were sequenced (GenBank Accession No. AY518517–AY518519). In a population derived from PI 96983 × Lee 68, one of the six clones, the *3gG2* gene, was co-segregating with *Rsv1* and the distance between *3gG2* and *Rsv1* was 0 cM; hence, *3gG2* (Wm82.a2.v1: *Glyma.13g190800*; Gm13: 30,426,359-30,430,201) became a strong candidate gene for the *Rsv1* locus. This 3390 bp gene encodes a protein product similar to non-N-terminal (non-TIR) NBS-LRR, common for disease resistance genes. In addition to PI 96983, Marshall (*Rsv1-m*) and Ogden (*Rsv1-t*) were reported to contain the *3gG2* gene [69,130].

After the reporting of the *3gG2* gene, it became the main target of searching for SMV resistance at the *Rsv1* locus. A PCR-based primer, Rsv1-f/r, was developed based on this *Rsv1* candidate gene. Shi et al. (2008b) [131] confirmed the presence of the *3gG2* in all genotypes carrying different *Rsv1* alleles (*Rsv1*, *Rsv1-h*, *Rsv1-k*, *Rsv1-m*, *Rsv1-n*, *Rsv1-r*, and *Rsv1-t*) except *Rsv1-y* (*Rsv1-s* was not tested), and the SSR marker Satt114 was found to be linked to Rsv1-f/r with a distance of 5.42 cM. Two allele-derived SNP markers were developed from the *3gG2* allele-specific primer pair 3gG2-f1/r1 between Essex and genotypes that contained the *3gG2* allele at the *Rsv1* locus, which proved to be useful in the detection of the *3gG2* gene in diverse soybean germplasm [132].

Yang et al. (2013) [133] concluded that there might be one or two dominant R-genes tightly flanking the *Rsv1* locus by performing a cross of PI 96983 (*Rsv1*) × ‘Nannong 1138-2’ (*rsv*) and screening their RILs with molecular markers. This study discovered the potential *Rsc-pm* locus which confers resistance to the Chinese strains SMV-SC3, SC6, and SC17, positioned between BARCSOYSSR_13_1128 (Wm82.a2. Gm13:30119784-30119825) and BARCSOYSSR_13_1136 (Gm13:30464888-30464941). Another gene, *Rsc-ps*, confers resistance to SMV-SC7, and was reported between BARCSOYSSR_13_1140 and BARCSOYSSR_13_1155 (Table 2). Recently, the *Rsv1*-*h* gene that confers resistance to the Chinese strains SC6-N and SC7-N was fine-mapped in cultivar Suweon 97 to a 97.5-kb region on chr. 13 (LG F) (29,815,195–29,912,667 bp) and was flanked by BARCSOYSSR_13_1114 and BARCSOYSSR_13_1115. Eight potential candidate genes were present in that region, of which *Glyma.13g184800* and *Glyma.13g184900* (Wm82.a2.v1) encode the *CC*-*NBS*-*LRR* type of gene [134]. The fine-mapping of the SMV-SC3 resistance gene in near-isogenic lines (NILs) derived from ‘Qihuang-1 (R)’ and Nannong 1138-2 (S) cross revealed a 180 kb region of chr. 13 (LG F) with 17 annotated genes. Transcriptome and expression analyses pinpointed four candidate genes: *Glyma.13g190000*, *Glyma.13g190300*, *Glyma.13g190400*, and *Glyma.13g190800* (Wm82.a2.v1) [135]. The resistance gene *Rsv1-r* in Raiden was fine-mapped to a region (~154.5 kb) between two SNP markers, SNP-38 and SNP-50, and two CC-NBS-LRR genes (*Glyma.13g184800* and *Glyma.13g184900*) (Wm82.a2.v1) exhibited significant divergence in SMV resistance between Raiden and Williams 82 [136].

## 7. Discovery and Rejection of *Rsv2* Locus

Based on a genetic analysis, the soybean line ‘OX670’ was identified as carrying a new SMV resistance gene independent of *Rsv1*. The 15R:1S segregation ratio observed in the F_2_ population of OX670 (R) × ‘L78-379’ (S) indicated that the gene in OX670 is independent of the *Rsv1* locus, and, therefore, the *Rsv2* symbol was assigned to SMV resistance in OX670 [145]. However, it was later presumed that this novel resistance gene was derived from Raiden [55,146]. Based on allelism tests of Raiden and L88-8431 (Williams isogenic line carrying the R-gene from Raiden), SMV resistance was confirmed to be at the *Rsv1* locus with partial dominance, and named as the *Rsv1-r* allele [55]. Later, it was documented that OX670 carries the two SMV resistance loci *Rsv1* and *Rsv3*, derived from Raiden and ‘Harosoy’ (PI 548573), respectively; therefore, the *Rsv2* in Raiden and its ancestors do not exist [5,60]. In another study, the reaction pattern to different SMV strains and genetic modes conferred by the gene from Raiden were similar to those conferred by the *Rsv1-s* allele in the breeding line LR1 derived from PI 486355 [37,110]. PI 486355 was shown to contain two resistance genes: *Rsv1-s* and *Rsv4*. Both alleles, *Rsv1-r* in Raiden and *Rsv1-s* in LR1, confer resistance to SMV-G1 through G4 and G7, but necrosis to G5 and G6 (Table 1). The SMV reaction pattern and the genetic behavior of *Rsv1-r* are like those of other *Rsv1* alleles. It is highly likely that the *Rsv1-r* in Raiden and *Rsv1-s* in PI 486355 originated from the same source. The *Rsv1-r* allele is gene-dosage dependent, because it confers resistance in the homozygous state and systemic necrosis in heterozygotes infected by SMV-G7 [37,55,110].

## 8. *Rsv3* Locus Confers Resistance to More Virulent SMV Strains

In contrast to *Rsv1*, alleles of *Rsv3* exhibit dominance and confer resistance to more virulent strains, G5 through G7, but susceptibility to less virulent strains, G1 through G4 (Table 1). Like *Rsv1*, *Rsv3* is strain-specific and associated with HR [113]. The *Rsv3* locus contains at least six alleles identified in OX686, Harosoy, L29, PI 61944, PI 61947, and PI 399091 [58,59,60,61,62].

*Rsv3* was first identified as an independent and dominant gene in OX686, conditioning stem-tip necrosis to SMV-G1 [103]. The OX686 line was derived from a ‘Columbia’ × Harosoy cross. Columbia was presumed to carry *Rsv3* and *Rsv4*, conferring resistance to all SMV strains [71], while Harosoy carries resistance to SMV-G5 to G7 at the *Rsv3* locus, but is susceptible to G1 through to G4. Therefore, it is most likely that *Rsv3* in OX686 was derived from Columbia rather than Harosoy. The *Rsv3* locus in Harosoy is partially dominant, giving rise to a necrotic reaction in heterozygous plants infected by SMV-G7 [60]. The pedigree analysis of Harosoy and Hardee indicated that the resistance gene *Rsv3* in both genotypes was derived from Chinese accessions, but their relationship is unknown. They showed the same reaction pattern to SMV strains, but different genetic behavior, being allelic at the same locus [60]. Buss et al. (1999) [58] reported that L29, a derivative of Hardee, exhibited resistance to G5, G6, and G7, but susceptibility to G1 through to G4 (Table 1). Inheritance and allelism studies showed that L29 carries a new single, completely dominant allele at the *Rsv3* locus. Based on the same strategy, PI 61944 was reported to contain a different, single, dominant *Rsv3* allele with a distinct symptom pattern of necrosis to SMV-G1 and resistance to G7 [61]. Shakiba et al. (2012) [62] reported two new alleles of the *Rsv3* locus, *Rsv3-c* in South Korean PI 399091, and *Rsv3-h* in a Chinese land race PI 61947. The *Rsv3-c* gives a susceptible reaction to G1, G2, and G6 strains, but resistance to G3, G5, and G7, whereas *Rsv3-h* confers necrosis to G1 and G2, and resistance to G3 to G7. The CI protein is found to be the effector of *Rsv3*-mediated resistance [18,147,148].

## 9. Mapping of the *Rsv3* Locus

The *Rsv3* gene was mapped on chr. 14 (LG B2) in two crosses of L29 (R) × Lee 68 (S), and Tousan 140 (R) × Lee 68 (S) using data collected from the F_2_ generations. Primarily, the *Rsv3* gene was flanked by A519F/R at 0.9 cM and M3Satt at 0.8 cM (Table 2). Additionally, the SSR marker Satt063 was mapped at the same side of A519 with a distance of 6.5 cM to *Rsv3* [138]. Later, the locus was mapped in the J05 soybean cultivar using an F_2_ population derived from a cross with Essex (S). Two SRR markers, Sat_424 and Satt726, were closely linked with *Rsv3* with a distance of 1.5 and 2.0 cM, respectively [69] (Table 2).

For fine-mapping, three populations, L29 × ‘Sowon’ (BC_3_F_2_), L29 × Lee 68 (F_2_), and Tousan 140 × Lee 68 (F_2_), were developed. The sequence region of 154 Kb between A519F/R and M3Satt revealed Coiled-Coil Nucleotide-Binding Leucine-Rich Repeat CC-NB-LRR *Rsv3* candidate genes (Wm82.a2.): *Glyma.14g204500*), *Glyma.14g204600*, *Glyma.14g205000*, and *Glyma.14g205300* [149] (Table 3). Wang et al. (2011) [139] crossed ‘Dabaima’ (R) × ‘Nannong1138-2’ (S) and mapped resistance to the Chinese strain SMV-SC4 into the 100 Kb interval between BARCSOYSSR_14_1413 and BARCSOYSSR_14_ 1416 at the *Rsv3* locus (Table 2). Quantitative real-time PCR analysis further identified *Glyma.14g204600*, *Glyma.14g205000*, and *Glyma.14g205200* (Wm82.a2.v1) to be likely involved in this resistance (Table 3). Redekar et al. (2016) [150] reported *Glyma*.*14 g204700*, as the *Rsv3* candidate gene, as being highly expressed after SMV inoculation. Recently, *Glyma.14g204700* was proposed as the *Rsv3* candidate gene based on transient expression in tobacco and homology modeling. This research revealed point mutations in susceptible lines, and a 30 bp deletion in resistant lines [151]. Another piece of research also reported that *Glyma.14g204700* in L29 is likely to be the *Rsv3* gene that confers strain-specific resistance to SMV by over-expression and the virus-induced transient silencing of this candidate gene in soybeans [152]. Based on the gene regulatory network using transcriptomic data, *Rsv3* causes a defense via a complex phytohormone network, where abscisic acid, cytokinin, jasmonic acid, and salicylic acid pathways are suppressed, including a transcription factor MYC2 encoded by *Glyma.07g051500* [153,154].

## 10. *Rsv4* Locus Confers Resistance to Most or All SMV Strains

Four alleles at the *Rsv4* locus have been identified in V94-5152, PI 88788, and ‘Beeson’ (PI 548510) (*Rsv4-b*) [5,57,63,71,160]. The *Rsv4* alleles confer complete dominant resistance to all SMV strains, but in some cases may show early resistance (ER) at the seedling stage and the expression of delayed mild susceptibility with mosaic symptoms at a late stage [5,63]. Unlike *Rsv1* and *Rsv3*, no hypersensitive reaction has been observed, suggesting that the *Rsv4* locus may have unique molecular mechanisms [5,71,113,156]. Due to a wide spectrum of resistance to SMV strains, there is great interest in pyramiding the *Rsv4* locus with *Rsv1* and *Rsv3* loci as a strategy to defend against infections caused by multiple SMV strains [32].

The first *Rsv4* allele was found in PI 486355 (*Rsv1Rsv4*) [37]. Both genes were isolated in F_3:4_-derived lines from the cross of PI 486355 (R) × Essex (S). The ‘D26’ soybean line, later named ‘LR2’, was advanced to F_6:7_ and released as V94-5152. V94-5152 carries a single *Rsv4* gene conferring resistance to all strains [37,63,70]. Later, another allele at the *Rsv4* locus was identified in PI 88788 [5]. This resistance was confirmed in crosses PI 88788 × LR2 and PI 88788 × V94-5152, and reported to be controlled by a single dominant *Rsv4* gene. Shakiba et al. (2011, 2013) [57,160] proposed a new *Rsv4-b* allele in the soybean cultivar Beeson, displaying a unique resistance pattern when compared with other *Rsv4* alleles: early resistance to strains G1, G2, and G6, resistance to G5 and G7, and susceptibility to G3 (Table 1). Recently, inheritance and allelic studies revealed that resistance to SMV in PI 438307 is controlled by a single dominant *Rsv4-v* allele. PI 438307 is resistant to SMV-G1 through to G6, and resistant at seedling stages to SMV-G7 [64]. The P3 protein was found to be the effector of *Rsv4*-mediated resistance [22,161].

## 11. Mapping of *Rsv4* Locus

The *Rsv4* locus was mapped on chr. 2 (LG D1b) using an F_2_ population derived from the cross V94-5152 (R) × Lee 68 (S). The locus was flanked between Satt542 at 4.7 cM and Satt558 at 7.8 cM (Table 2) [128,144]. Later, two expressed sequence tag (EST) markers, AI856415-g or AI856415-S and BF070293-S, were mapped at 2.8 cM on one side of the gene, and two EST markers AW307114A (3.3cM) and AW471852A (2.4 cM) on the other side [141]. In addition, Fu et al. (2006) [162] mapped resistance to the Chinese strain SMV-SC7 in ‘Kefeng No.1’ to a 2.65 Mb region utilizing Satt266, Satt634, Satt558, Satt157, and Satt698 linked to *Rsv4* with distances of 43.7, 18.1, 26.6, 36.4, and 37.9 cM, respectively (Table 2).

Several studies focusing on fine-mapping *Rsv4* have been reported. Saghai Maroof et al. (2010) [156] utilized the whole genome shotgun sequence for fine-mapping *Rsv4* in two populations derived from D26 (R) × Lee 68 (S) and V94-5152 (R) × Lee 68 (S). Six markers were used to localize the gene in the 1.3 cM region in both mapping populations with a physical interval of less than 100 Kb. In this region, several candidate genes, *Glyma.02g121200*, *Glyma.02g121300*, *Glyma.02g121400*, *Glyma.02g121500*, *Glyma.02g121600*, and *Glyma.02g121800* (Wm82.a1.v1), were proposed (Table 3). Wang et al. (2011) [155] analyzed populations derived from Kefeng No.1 (R) × Nannong 1138-2 (S) to map resistance to the SMV-SC8 strain. Two SSR markers, BARCSOYSSR_02_0610 and BARCSOYSSR_02_0616, flank the gene within a 200 Kb interval. Further, five candidate genes were determined by an expression analysis: *Glyma.02g120700*, *Glyma.02g120800*, *Glyma.02g121500*, *Glyma.02g121900*, and *Glyma.02g122000* (Wm82.a1.v1) (Table 3). In a study by Li et al. (2015) [159] based on the cross Kefeng No.1 (R) × Nannong 1138-2 (S), the *Rsc18A* locus (resistance to SMV-SC18) was mapped within an 80 Kb region, where six putative genes were predicted, and of which three (*Glyma02g127800*, *Glyma02g128000*, and *Glyma02g12820*) displayed differences at the amino acid level (Table 3). Additionally, Yan et al. (2015) [143] used a set of 191 soybean accessions for a genome-wide association study, and 184 RILs of Kefeng No.1 (R) × Nannong 1138-2 (S) to identify resistance to the SMV-SC7 strain. Among 19 SNPs detected via an association analysis, BARC-021625-04157 was located in the 2.65 Mb region, and fine-mapped to the *Rsv4* region of approximately 158 Kb between BARCSOYSSR_02_0621 and BARCSOYSSR_02_0632 (Table 2). From fifteen genes located in this region, three candidate genes (*Glyma.09g208900*, *Glyma.11g079900*, and *Glyma.16g159700*) (Wm82.a2.v1) were proposed (Table 3). Based on different confidence intervals, it was suggested that Kefeng No. 1 carries at least two different SMV-resistance genes near the *Rsv4* locus. Recently, the inheritance of SC5 resistance was analyzed in a Kefeng No.1 (R) × ‘NN1138-2’ (S) RIL population, and the gene was fine-mapped into a 500 kbp genomic region containing 38 putative genes. Furthermore, based on the integration of the SNP-phenotype association analysis and qRT-PCR expression analysis, *Glyma.02g122100* was proposed as the most possible candidate gene for SC5 resistance [158]. Ilut et al. (2016) [142] analyzed the BC_3_F_2_ population derived from the cross of V94-5152 (R) × Sowon (S) and delimited the *Rsv4* locus to the 120 kb region between markers Rat2 and S6ac. Two candidate genes, *Glyma.02g122000* and *Glyma.02g122100*, were proposed. Recently, Ishibashi et al. (2019) [157] used a cross of ‘Peking’ (R) × ‘Enrei’ (S) to map the *Rsv4* gene into a 9.8 kbp intergenic region just between genes *Glyma.02g122000* and *Glyma.02g122100*. In the SMV-susceptible soybean cultivar Enrei, this region contained two tandemly repeated homologous open reading frames (ORFs, NM_001249088 and NM_001253944), which encode RNase H-like proteins, whereas in the SMV resistance cultivar Peking, there was a 3.6 kbp deletion resulting in only one ORF. It was hypothesized that the encoded RNase H-like protein likely causes the degradation of viral dsRNA by entering into the membranous viral replication compartment.

## 12. Novel Locus *Rsv5*

*Rsv1-y* in the York soybean has been classified as an allele of the *Rsv1* locus; however, several phenomena recently raised a question of whether the *Rsv1-y* allele belongs to the *Rsv1* or if it is a distinct, but tightly linked, locus. According to the study by Shi et al. (2008b) [131], a PCR-based marker Rsv1-f/r for the detection of the *Rsv1* candidate gene *3gG2* (Wm82.a2.v1: *Glyma.13g190800*), completely linked to *Rsv1*, could amplify a specific sequence from soybean genotypes carrying various *Rsv1* alleles, except for *Rsv1-y*. Recently, Yang et al. (2013) [133] concluded that there might be one or two dominant R-genes tightly flanking the *Rsv1* locus by performing a cross of PI 96983 (R) × Nannong 1138-2 (S). The potential *Rsc-pm* gene confers resistance to the Chinese SMV strains SC3, SC6, and SC17, and was positioned between BARCSOYSSR_13_1128 and BARCSOYSSR_13_1136. The other gene, *Rsc-ps*, confers resistance to SMV-SC7, and was reported between BARCSOYSSR_13_1140 and BARCSOYSSR_13_1155 (Table 2). The *Rsv1* locus is located at the genomic region on the long arm of chr. 13 (LG F), where multiple R genes have previously been reported [121,130] and is tightly linked to a cluster of genes containing an NBS-LRR (www.soybase.org, accessed on 18 May 2022). This area of the chromosome is extremely complex, conferring resistance not only to SMV, but also to soybean aphids [163] and other plant pathogens, e.g., *Phytophthora* [164] and *Fusarium* [165]. The *Rsv1* locus seems to be complex itself, with the possibility of having a variety of at least ten different copies of the same gene. In a recent study by Klepadlo et al. (2017b) [65], two alleles of the *Rsv1* locus were analyzed by crossing PI 96983 (*Rsv1*) and York (*Rsv1-y*) to determine whether *Rsv1* and *Rsv1-y* belong to the same locus. The occurrence of segregating and susceptible lines indicated tight linkage between the two SMV R-genes, with an estimated genetic distance of 2.2 cM. A new symbol of *Rsv5* was assigned for the resistance gene in York to replace the original *Rsv1-y* allele name [65]. This locus separation helps explain why York is susceptible to G5-7, a rare pattern for *Rsv1* alleles. The P3 protein has been considered as a possible effector of *Rsv5*-mediated resistance [21].

## 13. Gene Combinations Reduce Vulnerability of SMV Infection

The inheritance of SMV resistance has been extensively studied for many years. In over 80% of reported cultivars, resistance is controlled by a single gene [31,37,94,146,166]. Only a few soybean accessions carry two resistance genes in different combinations *(Rsv1Rsv3*, *Rsv1Rsv4*, or *Rsv3Rsv4)* that interact in a complementary fashion, giving resistance to SMV-G1 through G7 (Table 1). Up to now, only one soybean genotype ‘8101’ has been found to carry three resistance genes, giving rise to resistance to all SMV strains [72].

Eight genotypes, namely, OX670, ‘Hourei’, Tousan 140, J05, ‘Jindou 1’, Zao 18, PI 486355, and Columbia, carry two resistance genes in various combinations of *Rsv1*, *Rsv3*, and *Rsv4* [60,66,67,68,69,71]. The presence of two genes for SMV resistance substantially reduces plant vulnerability to infection by various strains/isolates [70,89]. The *Rsv1* allele in OX670 is derived from Raiden and the *Rsv3* allele from Harosoy [60]. The *Rsv1* in Tousan 140 was similar to *Rsv1-y* for the same resistance reaction to G1 and G3, local necrotic lesion to G2, and susceptibility to G5 through G7, and *Rsv3* for the same resistance to G5-G7 and susceptibility to G1-G3 [66]. The *Rsv1* and *Rsv3* in Hourei were similar to the two genes in Tousan 140 that condition similar reactions to G1 and G7, but sources of the two genes are not known. *Rsv1* allele in Tousan 140 gives resistance to G1 and G7 strains, whereas *Rsv3* allele confers resistance to G7, but susceptibility to G1 [66]. *Rsv1* in Zao 18 is similar to *Rsv1-y* (*Rsv5*) conditioning resistance to G1 and susceptibility to G7, and *Rsv3* conferring resistance to G7 and susceptibility to G1 [68]. The soybean cultivar J05 was also identified carrying *Rsv1* and *Rsv3* [67,69], but it is not known whether or not the two genes are new alleles at *Rsv1* or *Rsv3*. The digenic combination of *Rsv1* and *Rsv3* provides a wide potential for SMV resistance. Jindou, a soybean cultivar developed in China, is resistant to SMV-G1 through to G7, and was identified to contain resistance genes of *Rsv1-y* (*Rsv5*) and *Rsv3* [167].

Only one soybean accession, PI 486355, was reported to carry *Rsv1-s* and *Rsv4* [37,63,70]. PI 486355, originally named SS74185 and collected in South Korea, is resistant to SMV G1 through to G7 in the US and 23 SMV isolates from China [50]. *Rsv1-s* confers resistance with partial dominance to G1-G4 and G7, but necrosis to G5 and G6, which is similar to the reaction pattern of *Rsv1-r* in Raiden. In contrast, *Rsv4* exhibited complete dominance and confers resistance to all SMV strains (G1-G7). These two genes were separated in two different F-_3:4_ derived lines, LR1 and LR2, from a cross between PI 486355 (R) and Essex (S) [37].

Columbia contains two dominant genes, *Rsv3* and *Rsv4*, and displays resistance to all SMV strains [59,71]. In populations of Columbia × Lee 68, a unique response to SMV-G1 has been designated to be resistant at the early seedling stage with delayed mild mosaic symptoms approximately three weeks after initial inoculation. The same populations remained resistant to SMV-G7. The *Rsv3* in Columbia and OX686 is the same gene that confers resistance to G7, but necrosis to G1. The *Rsv4* gene in Columbia confers resistance to G7, and *Rsv3* and *Rsv4* interact in a complementary fashion and provide complete resistance to G1–G3 and G5–G7, and stem-tip necrosis to G4 [71].

## 14. Other SMV Gene Definitions

In addition to four SMV resistance loci (*Rsv1*, *Rsv3*, *Rsv4*, and *Rsv5*) and their alleles, several other genes, *Rsa*, *Rn1*, *Rn3*, *Rsc7*, *Rsc8*, *Rsc9*, and *Rsc13* have been mapped on chr. 2 (LG D1b), and *Rsc14* on chr. 13 (LG F) for resistance to Chinese SMV strains (Table 4). Due to differences in SMV strain classification systems between the US and China, the resistance loci identified in both countries cannot be generalized under the same names, and the relationship among these loci is unknown. Likely, *Rsc14* shares the same locus of *Rsv1*, whereas *Rsa*, *Rn1*, *Rn3*, *Rsc7*, *Rsc8*, *Rsc9*, and *Rsc13* share the same locus as *Rsv4* (Table 4). It is necessary to conduct allelism tests to determine whether the eight genes identified in China were located at the same loci as *Rsv1*, *Rsv3*, and *Rsv4* reported in the US using the same SMV strains.

Zhang et al. (1999) [176] reported a single dominant gene, *Rsa*, in the soybean cultivar Kefeng No.1 conferring resistance to the Chinese SMV-Sa strain, using a cross of Kefeng No. 1 (R) × Nannong 1138-2 (S) and the bulk segregant analysis (BSA) method. Two RAPD markers, OPW-05_660_ and OPAS-06_1800_, were found to be linked to this gene with genetic distances of 10.1 and 22.2 cM. Recombinant inbred lines of the same cross were utilized by Wang et al. (2004) [177] to study genetic resistance in the Kefeng No.1 cultivar to five Chinese strains: SMV-Sa, SC8, SC9, N1, and N3. The results indicated that the resistance to each of the five strains was controlled by a single, separate but linked, dominant gene. *Rsa* was found to be linked to *Rn1*, *Rn3*, and *Rsc9* with 21.4, 23.5, and 35.3 cM distances, respectively, while *Rsc8* was found to be linked only to the *Rn1* locus with a 35.8 cM distance on chr. 2 (LG D1b). RFLP markers A691T, K4771, and LC5T were linked to *Rn1* and *Rn3* with distances of 15.04, 17.82, and 15.37, and 16.14, 17.82, and 16.58 cM, respectively. Fu et al. (2006) [162] identified the *Rsc7* locus in Kefeng No.1 for resistance to Chinese SMV-SC7 in F_2_ and RIL populations derived from a cross of Kefeng No.1 × Nannong 1138-2. The *Rsc7* was also mapped on chr. 2 (LG D1b) with five SSR markers, Satt266, Satt634, Satt558, Satt157, and Satt698, linked with distances of 43.7, 18.1, 26.6, 36.4, and 37.9 cM, respectively, and flanking markers of BARCSOYSSR_02-0621 and BARCSOYSSR_02-0632 [143]. However, a linkage analysis indicated that *Rsc7* was not linked to *Rsc8* and *Rsc9* [178]. Later, using an RIL population derived from the cross Kefeng No.1 × Nannong 1138-2, seven resistance genes *Rsa*, *Rn1*, *Rn3*, *Rsc7*, *Rsc8*, *Rsc9*, and *Rsc13* were mapped on chr. 2 MLG D1b [179]. The *Rsc8* was fine-mapped in the F_2_ population from the cross between Kefeng No.1 and Nannong 1138-2 using flanking markers of BARCSOYSSR_02_0610 and BARCSOYSSR_02_0616, and several candidate genes were identified, including *Glyma.02g120700*, *Glyma.02g120800*, *Glyma.02g121500*, *Glyma.02g121900*, and *Glyma.02g122000* (Wm82.a2.v1) [174]. Based on another recent study by Luan et al. (2020) [173], *Rsc13* was fine-mapped in an RIL population derived from Kefend No.1 × Nannong 1138-2 using two BARCSOYSSR_02_0610 and BARCSOYSSR_02_0621, and 11 candidate genes were identified (*Glyma.02g13361*, *Glyma.02g13371*, *Glyma.02g13380*, *Glyma.02g13401*, *Glyma.02g13420*, *Glyma.02g13450*, *Glyma.02g13460*, *Glyma.02g13470*, *Glyma.02g13495*, *Glyma.02g13520*, and *Glyma.02g13530*). These genes reported in China could be alleles at the *Rsv 4* locus identified in the US. Recently, Shen et al. (2022) [180] reported *Rsc9* in a region of approximately 163 kb on chr. 2 (LG D1b) by fine-mapping in F_2_ and RIL populations of Kefeng No.1 × Nannong 1138-2 using two flanking SSR markers (BARCSOYSSR_02_0610 and BARCSOYSSR_02_0618) and concluded that *Glyma.02g122000* and LOC100812666 may be associated with the soybean resistance to SMV SC9.

Li et al. (2006) [181] mapped the *Rsc14* locus on chr. 13 (LG F) linked to *Rsv1* with a distance of 5.3 cM in an F_2_ population from PI 96983 (R) × Nannong 1138-2 (S). Five SSR markers, Sat_297, Sat_234, Sat_154, Sct_033, and Sat_120, were found closely linked to *Rsc14*, with genetic distances of 14.5, 11.3, 4.3, 3.2, and 6 cM, respectively. Another gene, designated as *Rsc14Q*, was also mapped on chr. 13 (LG F) in the F_2_ population of Qihuang No.1 (R) × Nannong 1138-2 (S), and three SSR markers, Sat_234, Satt334, and Sct_033, were found to be linked with genetic distances of 7.2, 1.4, and 2.8 cM, respectively. It was suggested that *Rsc14* and *Rsc14Q* might be at the same locus because both were located between Sat_234 and Sct_033 markers, with distances from Sct_033 at the same side being 3.2 and 2.8 cM, respectively. In essence, *Rsc14* and *Rsc14Q* found in China may be alleles of *Rsv1* or *Rsv5* identified in the US.

Recently, Yin et al. (2021) [182] reported *Rsc4-3*, located on chr. 14, to mediate resistance to multiple SMV strains such as SC3, SC7, SC8, SC11, SC14, G5, and G7. This gene encodes a cell-wall-localized NLR-type resistant protein. The cylindrical inclusion (CI) protein is the avirulent gene for *Rsc4-3*-mediated resistance, and partially localizes to the cell wall and interacts with *Rsc4-3*.

A significant genetic source of SMV resistance was also mapped on chr. 6 (LG C2). Yang and Gai (2011) [175] crossed ‘RN-9’ (R) × ‘7605’ (S) to study the resistance to the Chinese SMV-SC15 strain. Results indicated that a single dominant gene, designated as *Rsc15*, conferred the SMV resistance, and it was located between Sat_213 and Sat_286 with distances of 8.0 and 6.6 cM, respectively. The candidate gene *Glyma.18g154900* in RN-9 encodes the cytochrome protein GmCYB5 targeting the P3 protein of SMV to inhibit its proliferation (Luan et al., 2019) [183]. This novel resistance gene has never been found in germplasm infected by American SMV strains. Thus, this gene has not been found or reported in the US germplasm.

## 15. Genetic Resources and Breeding for SMV Resistance

A set of NILs is a powerful genetic resource for the identification of genes or QTLs associated with a trait of interest. Each NIL carries a genomic segment associated with the trait of interest from a donor parent and is incorporated into the genomic background of the recurrent parent. Constructing a set of NILs requires a bi-parental hybridization followed by repetitive backcrossing and an extensive molecular study [184]. Two sets of isogenic lines for SMV resistance alleles were constructed (Table 5). Williams isogenic lines (L-series) were developed by Richard L. Bernard, University of Illinois, and were derived from backcrossing Williams (S) with 10 different resistant lines resulting in isogenic lines carrying different alleles of the *Rsv1* and *Rsv3* loci [118]. Essex isogenic lines (V-series) carrying different alleles of *Rsv1*, *Rsv3*, and *Rsv4* loci were developed by Glenn Buss, Dep. of Crop and Soil Environmental Sciences, Virginia Polytechnic Institute and State University. Essex isogenic lines showed resistant, necrotic, or susceptible reactions when infected by the same SMV strain from G1 to G7. For example, infection by SMV-G1 provided resistance in ‘V94-3971’ and ‘V97-9003’, necrosis in ‘V262’, and susceptibility in ‘V229’. In this case, induced symptoms did not depend on the virus strain, but instead on the host genotype [63,113,118]. In contrast, a specific genotype could exhibit multiple symptoms when infected with different strains. For example, V94-3971 shows resistance to G1 but necrosis to G7, V229 exhibits susceptibility to G1 but resistance to G7, and V262 is necrotic to G1 but susceptible to G7. These isolines with different SMV resistance alleles in the same genetic backgrounds could allow for future investigations on specific resistance gene–SMV strain interactions. These isolines may also be useful in future studies of the gene-for-gene concept in the soybean–SMV system.

Several gene-pyramided lines were developed utilizing MAS for the incorporation of SMV resistance genes at different loci to develop cultivars with durable resistance [89]. Saghai Maroof et al. (2008) [113] pyramided *Rsv1*, *Rsv3*, and *Rsv4* using three different Essex isogenic lines, each carrying a single R-gene. Two-gene and three-gene isogenic lines with combinations of *Rsv1Rsv3*, *Rsv1Rsv4*, and *Rsv1Rsv3Rsv4* acted in a complementary manner conferring resistance against all SMV strains, whereas isogenic lines of *Rsv3Rsv4* displayed a late susceptible reaction to the selected SMV strains not observed in the donor sources. Shi et al. (2009) [89] also pyramided the same genes from the cross J05 (*Rsv1Rsv3*) × V94-5152 (*Rsv4*) using eight markers: Sat_154, Satt510, and Rsv1-f/r for *Rsv1*; Satt560 and Satt063 for *Rsv3*; and Satt266, AI856415, and AI856415-g for *Rsv4*. Five F_4:5_ lines were identified to be homozygous for all eight marker alleles and presumably carried all three SMV resistance genes that potentially provide multiple and durable resistance to SMV. Cervantes (2012) [185] performed the cross PI 96983 (*Rsv1*) × Columbia (*Rsv3Rsv4*) and developed 11 pyramided lines with three resistance genes homozygous and heterozygous on each locus. Three SSR markers, Satt510, Sct_064, and Satt296, were mainly used for screening *Rsv1*, *Rsv3*, and *Rsv4* loci. Three SMV resistance genes, *Rsc4*, *Rsc8*, and *Rsc14Q*, were identified and mapped on soybean chromosomes 14, 2, and 13 from Dabaima, Kefeng 1, and Qihuang 1 cultivars, respectively [186]. The soybean cultivar Nannong 1138-2 is widely grown in the Yangtze River valley of China. The crosses were performed between (Qihuang 1 × Kefeng 1) and (Dabaima × Nannong 1138-2). Ten SSR markers linked to these three resistance loci were used for gene pyramiding, and the progenies were evaluated by inoculations with 21 SMV strains from China. Five F7 homozygous pyramided families exhibited resistance to all 21 strains [186].

SMV resistance was also successfully introgressed into high-yielding backgrounds in breeding programs. Harosoy has been used as a donor of resistance to Japanese strains SMV-C and D, resulting in resistant varieties such as ‘Fukuibuki’. Harosoy harbors the *Rsv3* gene conferring resistance to US SMV-G5 through to G7, and it was suggested that *Rsv3* confers resistance to the Japanese SMV-C and D. This resistance was introgressed by the recurrent backcrossing of Fukuibuki (R) × ‘Ohsuzu’ (S, high yield) with the assistance of MAS. Three years of field trials showed that the SVM-resistant breeding line ‘Tohoku 169’ was equivalent to Ohsuzu for important agronomic traits, such as seed size, maturity date, and yield [187].

Recently, there have been several transgenic soybean lines with SMV resistance that have been developed by overexpressing resistance-related genes (GmKR3 and GmAKT2) or expressing viral nucleic acid sequences of genetic elements (P1, CP, and HC-pro genes) for pathogen-derived resistance (PRD) [188,189,190,191,192,193,194]. Additionally, an RNAi technique has been successfully used to develop transgenic soybean lines with SMV resistance [189,195,196,197]. Specifically, a recent study conducted by Gao et al. (2019) [197] used a soybean endogenous gene, *eIF4E*, which is a major susceptibility factor for some RNA viruses in the RNAi technique. It confirmed that transgenic soybean lines with the silencing of the eIF4E gene had an enhanced and broad resistance to SMV-SC3, SC7, SC15, SC18, and SMV-R. In addition, transgenic soybean lines expressing P3 and HC-Pro genes also demonstrated a strong and stable resistance to SMV-SC3, SC7, SC15, SC18, and SMV-R, with a significant yield potential [195,196].

## 16. Presence and Future of SMV Resistance Research

The soybean is one of the most important commodity crops grown on a large scale around the world. Given the historical perspective of research on the physiological and genetic characteristics of disease caused by SMV, it is fundamental to look at the achievements of the past century, from the disease discovery in 1915, followed by the beginning of genetic studies in 1979, to its present status of genetic mapping and gene pyramiding. Great progress has been achieved in the past 35 years in our understanding of genetics and the molecular basis of resistance mechanisms. These discoveries have been advanced considerably by the whole-genome shotgun sequencing of Williams 82 [198]. During the earlier years of genetic studies, we gained much information on SMV resistance genes and their correlations with visible symptoms. Based on distinct reaction patterns of a given genotype to specific SMV strains, we can predict the host resistance gene or alleles (Table 1). For example, with resistance to SMV-G1 and mosaic or necrosis to G7, a given soybean accession may carry an R-gene at the *Rsv1* locus. In contrast, with a G1-susceptible and G7-resistant reaction, a genotype in question may carry an *Rsv3* allele. However, if a given accession shows resistance to all SMV strains, it may carry one of several gene possibilities, such as *Rsv1-h*, *Rsv4*, *Rsv1Rsv3*, *Rsv1Rsv4*, *Rsv3Rsv4*, or *Rsv1Rsv3Rsv4*, and in such cases, allelism and inheritance studies are necessary for the identification of resistance and the confirmation of allelomorphic relationships.

The identification of SMV resistance in diverse soybean accessions is crucial for breeding and production purposes, and the discovery of new resistance genes is likely to continue to provide effective SMV resistance to a broad and ever-changing range of SMV isolates. Although SMV resistance loci have been reported in many soybean genotypes, most of the modern commercial cultivars are susceptible to SMV, particularly to more virulent strains [34,125]. The use of genetic resistance is the primary method of controlling the disease, and with the emergence of new resistance-breaking SMV isolates, the search for new sources of host resistance is necessary to avoid potential vulnerability. Isogenic lines carrying different alleles at the same locus and mixing them up as a synthetic variety based on the distribution changes of SMV strains can be an effective approach to control multiple viral strains. Isogenic and pyramiding lines are also good materials for genetic analysis, gene mapping, gene expression, and cloning.

None of the SMV resistance genes have been cloned yet; however, mapped and fine-mapped regions pinpoint candidate genes that could be transformed soon. Genetic mapping studies have provided information on the approximate location of the genes. The *Rsv1* and *Rsv5* loci are difficult to map due to the presence of many tightly linked genes in the same genomic region that confer resistance to several other diseases, requiring new techniques to detect and map single genes. The identification of the exact locus is likely to begin a new era of detailed biology research, especially proteomics, to gain knowledge about molecular mechanisms that underlie soybean resistance to SMV. Fine-mapping, together with the availability of the complete soybean genome sequence, is likely to make the molecular cloning of these resistance genes possible, which could largely accelerate breeding programs for the control of SMV using these natural resistance sources.

Recent studies suggest that novel resistance to viral pathogens may be developed through advanced biotechnology [199]. Like other positive-sense RNA viruses, SMV requires many host factors to establish its infection in soybeans. The silencing or mutation of a host factor gene may generate SMV resistance. The recently developed precise genome-editing technology could make this approach accessible to the soybean industry and consumers as it becomes possible to develop transgene-free resistance to SMV. Target host factor genes can be precisely deleted through genetic transformation, followed by transgene removal via traditional breeding. This promise propels ongoing research to characterize molecular SMV–soybean interactions and identify host factors that are essential for SMV infection.

In summary, multiple SMV strains and isolates occur worldwide, causing various reactions in soybeans, including resistance, necrosis, and susceptibility. Resistance is a typical immune reaction without virus entry, multiplication, and movement, resulting in no symptoms of infection or disease. Necrosis could be localized or systemic with limited viral replication and movement, leading to very low levels of detectable virus. Necrosis is characterized as local necrotic spots, vine necrosis, stem browning, defoliation, and stem-tip necrosis, often leading to plant death. Susceptible reactions show typical mosaic, vein clearing, and leaf curling symptoms, with maximum viral replication and movement. SMV strains and isolates are classified into strain groups or pathotypes with different virulence and symptomology on host genotypes with different resistance genes and alleles. There are several classification systems in different countries, where different SMV strains/isolates and differential genotypes are used. Therefore, research is needed to compare and unify different classifications around the world. Genetic resources for SMV resistance are readily available and quite diverse. Several major resistance loci and many alleles have been identified, all of which reside in four chromosomal regions, namely, chrs. 2 (LG D1b), 6 (LG C2), 13 (LG F), and 14 (LG B2). *Rsv1* is the most common, with the most diverse alleles in soybean germplasm, predominantly found in the US gene pool. *Rsv1* alleles confer resistance with incomplete dominance to less virulent strains but necrotic or susceptible reactions to more virulent strains. In contrast, *Rsv3* alleles confer resistance to more virulent strains but susceptible reactions to less virulent strains. *Rsv4* confers resistance with complete dominance to all SMV strains only at the early stage. *Rsv1-h* is the single, most effective gene for SMV resistance. However, multiple allelic combinations such as *Rsv1Rsv3*, *Rsv1Rsv4*, and *Rsv1Rsv3Rsv4* are available, although rare, naturally in soybean germplasm, and can be achieved through gene pyramiding. Such gene combinations could provide multiple and durable resistance and prevent resistance from breaking down due to the genetic mutation of existing strains or the emergence of new isolates. *Rsv5* is a newly discovered locus and tightly linked to *Rsv1* in a common region on chromosome 13, where there are many disease resistance genes. The novel gene for SMV resistance on chr. 6 (LG C2) deserves more attention in future research. As with the strain differential systems in different countries, various resistance genes reported from different countries should be tested for allelism or differentiated through molecular approaches. To date, all the SMV resistance genes worldwide to date likely belong to five genetic loci: *Rsv1* and *Rsv5* on chr. 13 (LG F), *Rsv3* on chr. 14 (LG B2), *Rsv4* on chr. 2 (LG D1b), and the novel locus *Rsc15* on chr. 6 (LG C2). Molecular markers, particularly gene-specific markers, are very useful in the identification, differentiation, and selection of specific resistance genes/alleles of interest in germplasm screening, gene-pyramiding, or breeding and selection for SMV resistance. As the virus constantly evolves with changes of virulence and pathogenicity and possibilities of breaking resistance in the host, collaboration is needed between geneticists and virologists to understand the molecular mechanisms of interactions between the virus and host in preparation for strain mutation and resistance breaking. Breeders need to stay alert and leverage genetic information and technology advancements to take advantage of genetic resources and molecular tools in their breeding effort for resistance, as multiple, durable, and preventable resistance is the ultimate goal.

## Figures and Tables

**Figure 1 viruses-14-01122-f001:**
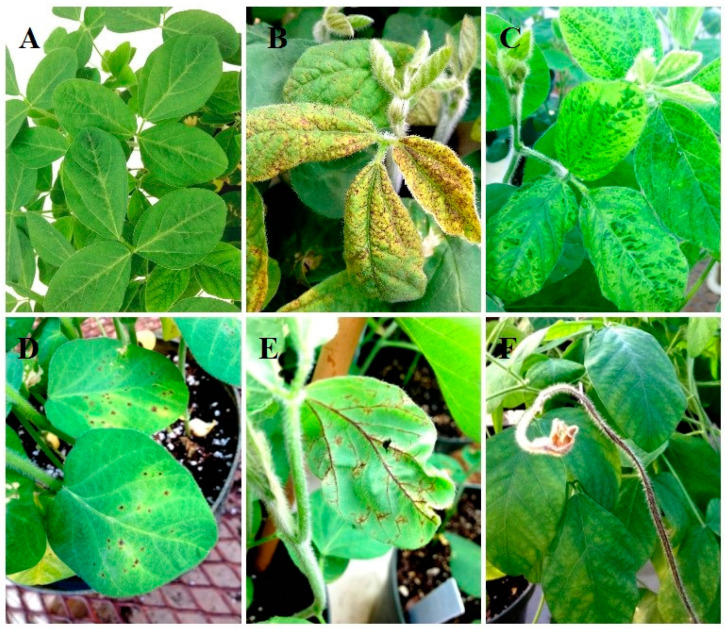
Symptoms of soybean genotypes infected with *Soybean mosaic virus*: (**A**), resistant genotype showing no symptoms of the disease; (**B**), necrotic genotype showing stem-tip necrosis; (**C**), susceptible genotype displaying mosaic symptoms; (**D**), local necrosis restricted to the infection site; (**E**), systemic necrosis with browning of leaf veins; (**F**), stem-tip necrosis with browning of the stem.

**Table 1 viruses-14-01122-t001:** Differential reactions of soybean genotypes to *Soybean mosaic virus* SMV strains in the US.

Genotype	Gene/Allele	Reaction to SMV Strain ^†^							Reference
		G1	G2	G3	G4	G5	G6	G7	
Essex/Lee68	*rsv*	S	S	S	S	S	S	S	[31]
PI 96983	*Rsv1*	R	R	R	R	R	R	N	[53]
Suweon 97	*Rsv1-h*	R	R	R	R	R	R	R	[54]
Raiden	*Rsv1-r*	R	R	R	R	N	N	R	[55]
Kwanggyo	*Rsv1-k*	R	R	R	R	N	N	N	[31]
Ogden	*Rsv1-t*	R	R	N	R	R	R	N	[31]
Marshall	*Rsv1-m*	R	N	N	R	R	N	N	[31]
PI 507389	*Rsv1-n*	N	N	S	S	N	N	S	[56]
LR1	*Rsv1-s*	R	R	R	R	N	N	R	[37]
Corsica	*Rsv1-c*	S	ER	S	-	ER	S	ER	[57]
L29	*Rsv3*	S	S	S	S	R	R	R	[58]
OX 686	*Rsv3*	N	N	N	N	R	R	R	[59]
Harosoy	*Rsv3*	S	S	S	S	R	R	R	[60]
PI 61944	*Rsv3-n*	N/S	N/S	R	-	R	R	R	[61]
PI 61947	*Rsv3-h*	N/S	N/S	R/N	-	R	R	R	[62]
PI 399091	*Rsv3-c*	S	S	ER	-	R	S	ER	[62]
V94-5152	*Rsv4*	ER	ER	ER	ER	ER	ER	ER	[63]
PI 88788	*Rsv4*	ER	ER	ER	ER	ER	ER	ER	[5]
Beeson	*Rsv4-b*	ER	ER	S	-	R	ER	R	[57]
PI 438307	*Rsv4-v*	R	R	R	R	R	R	ER	[64]
York	*Rsv5*	R	R	R	N	S	S	S	[65]
Hourei	*Rsv1Rsv3*	R	R	R	R	R	R	R	[66]
OX 670	*Rsv1Rsv3*	R	R	R	R	R	R	R	[60]
Tousan 140	*Rsv1Rsv3*	R	R	R	R	R	R	R	[66]
J05	*Rsv1Rsv3*	R	R	R	R	R	R	R	[67]
Zao18	*Rsv1Rsv3*	R	R	R	R	R	R	R	[68]
Jindou 1	*Rsv1Rsv3*	R	R	R	R	R	R	R	[69]
PI 486355	*Rsv1Rsv4*	R	R	R	R	R	R	R	[70]
Columbia	*Rsv3Rsv4*	R	R	R	R	R	R	R	[71]
8101	*Rsv1Rsv3Rsv4*	R	R	R	R	R	R	R	[72]

^†^ R, resistant (symptomless); ER, early resistant at seedling stage; N, necrotic (stem-tip or systemic necrosis); S, susceptible (mosaic); N/S, R/N, mixed reactions.

**Table 2 viruses-14-01122-t002:** PCR-based markers and their positions in relation to SMV resistance loci.

Locus	Marker	Marker/Interval Physical Position (Wm82.a2)	Type	Reference
*Rsv1*	Sat_297	Gm13:26976116-26976161	SSR	[91]
	Sat_234	Gm13:27656933-27657028	SSR	[91]
	Sat_154	Gm13:28506124-28506173	SSR	[69]
	Satt114	Gm13:28912878-28912928	SSR	[91]
	SOYHSP179	Gm13:29041580-29041694	SSR	[121]
	BARCSOYSSR_13_1114	Gm13:29815342-29815386	SSR	[134]
	3gG2-snp1	Gm13:29877164	SNP	[132]
	BARCSOYSSR_13_1115	Gm13:29912429-29912454	SSR	[134]
	BARCSOYSSR_13_1128	Gm13:30119784-30119825	SSR	[133]
	3gG2-snp2	Gm13:30402642	SNP	[132]
	3gG2-f/r	Gm13:30426359-30430201	GS	[131]
	BARCSOYSSR_13_1136	Gm13:30464888-30464941	SSR	[133]
	BARCSOYSSR_13_1138	Gm13:30472334-30472401	SSR	[137]
	BARCSOYSSR_13_1139	Gm13:30475113-30475164	SSR	[137]
	BARCSOYSSR_13_1140	Gm13:30501849-30501881	SSR	[133]
	BARCSOYSSR_13_1155	Gm13:30880128-30880147	SSR	[133]
	N11PF-snp2	Gm13:31012220	SNP	[132]
	Satt510	Gm13:31802616-31802642	SSR	[120]
	Sat_317	Gm13:32196864-32196911	SSR	[91]
	Barc-015435-01966	Gm13:32607605	SNP	[132]
*Rsv3*	Barc-012953-00413	Gm14:45086977		[132]
	Satt063	Gm14:45993741-45993800	SSR	[138]
	Sat_424	Gm14:46281881-46281928	SSR	[91]
	Satt726	Gm14:46326537-46326778	SSR	[91]
	A519-snp2	Gm14:46937343-46937343	SNP	[132]
	A519-snp4	Gm14:46937465-46937465	SNP	[132]
	A519-f/r	Gm14:46937953-46937958	In/Del	[138]
	BARCSOYSSR_14_1413	Gm14:46944330-46944387	SSR	[139]
	BARCSOYSSR_14_1416	Gm14:47007588-47007611	SSR	[139]
	M3aSatt	Gm14:47090758-47091111	SSR	[138]
	Satt560	Gm14:47170110-47170387	SSR	[140]
	Satt687	Gm14:48365113-48365133	SSR	[91]
*Rsv4*	Barc-011147-00855	Gm02:8380603	SNP	[132]
	Barc-025955-05182	Gm02:9689348	SNP	[132]
	Satt558	Gm02:10619724-10619771	SSR	[128]
	Sat_254	Gm02:11168955-11169036	SSR	[141]
	BF070293	Gm02:11314396-11314416	SSR	[140]
	AI856415	Gm02:11326591-11326637	SSR	[91]
	ss244712652	Gm02:11693604	SNP	[111]
	ss244712653	Gm02:11693900	SNP	[111]
	ss244712671	Gm02:11697977	SNP	[111]
	Satt634	Gm02:11778567-11778605	SSR	[91]
	BARCSOYSSR_02_0610	Gm02:11964524-11964549	SSR	[139]
	Rat2	Gm02:12044285	SSR	[142]
	BARCSOYSSR_02_0616	Gm02:12070465-12070492	SSR	[139]
	BARCSOYSSR_02_0621	Gm02:12156250-12156307	SSR	[143]
	Sms1	Gm02:12156384-12156096	SSR	[142]
	S6ac	Gm02:12163671-12163994	SSR	[142]
	BARCSOYSSR_02_0632	Gm02:12313357-12313394	SSR	[143]
	AW307114-indel	Gm02:12585482-12585482	In/Del	[132]
	BARC-021625-04157	Gm02:12623066	SNP	[143]
	Satt542	Gm02:13316465-13316521	SSR	[144]
	Satt296	Gm02:13335846-13335908	SSR	[140]
	Satt266	Gm02:14288260-14288310	SSR	[144]
*Rsv5*	Satt114	Gm13:28912878-28912928	SSR	[65]

**Table 3 viruses-14-01122-t003:** Summary of reported SMV resistance candidate genes at *Rsv1*, *Rsv3*, and *Rsv4* loci.

Locus	Wm82.a2.v1 ^†^	Wm82.a1.v1 ^†^	Physical Position ^‡^	Function ^§^	Reference
*Rsv1*	*Glyma.13g184800*	*Glyma13g* *25420*	Gm13:29858000-29863047	LRR Kinase	[134,136]
	*Glyma.13g184900*	*Glyma13g* *25440*	Gm13:29873181-29877088	LRR Kinase	[134,136]
	*Glyma.13g187600*	*Glyma13g25730*	Gm13:30134637-30143817	LRR Kinase	[133]
	*Glyma.13g187900*	*Glyma13g25750*	Gm13:30174410-30180072	LRR Kinase	[133]
	*Glyma.13g190000*	*Glyma13g25920*	Gm13:30355157-30359208	LRR Kinase	[135]
	*Glyma.13g190300*	*Glyma13g25950*	Gm13:30388583-30392233	LRR Kinase	[133,135]
	*Glyma.13g190400*	*Glyma13g25970*	Gm13:30402029-30409606	LRR Kinase	[133,135]
	*Glyma.13g190800*	*Glyma13g26000* *	Gm13:30423894-30430435	LRR Kinase	[130,131,133,135]
	*Glyma.13g191400*	*Glyma13g26070* **	Gm13:30472853-30474291	Sulfotransferase 1	[137]
*Rsv3*	*Glyma.14g204500*	*Glyma14g38500*	Gm14:46946496-46957734	LRR Kinase	[149]
	*Glyma.14g204600*	*Glyma14g38510*	Gm14:46968705-46974585	LRR Kinase	[149,155]
	*Glyma.14g204700*	*Glyma14g38533*	Gm14:46981104-46996696	LRR Kinase	[150,151,152]
	*Glyma.14g205000*	*Glyma14g38560*	Gm14:47005574-47019661	LRR Kinase	[149,155]
	*Glyma.14g205200*	*Glyma14g38580*	Gm14:47041931-47046048	Cytochrome P450	[155]
	*Glyma.14g205300*	*Glyma14g38590*	Gm14:47046209-47056610	LRR Kinase	[149]
*Rsv4*	*Glyma.02g120700*	*Glyma02g13310*	Gm02:11904074-11910578	Cytochrome P450	[139]
	*Glyma.02g120800*	*Glyma02g13320*	Gm02:11926840-11931251	LRR Kinase	[139]
	*Glyma.02g121400*	*Glyma02g13380*	Gm02:12028928-12030693	Unknown	[156]
	*Glyma.02g121500*	*Glyma02g13400*	Gm02:12065640-12082937	MADS Box TF	[139,142,156]
	*Glyma.02g121600*	*Glyma02g13420*	Gm02:12084714-12089043	MADS Box TF	[142,156]
	*Glyma.02g121700*	-	Gm02:12093068-12095722	Zinc-Finger	[142]
	*Glyma.02g121800*	*Glyma02g13450*	Gm02:12106073-12107969	Stress Protein	[142,156]
	*Glyma.02g121900*	*Glyma02g13460*	Gm02:12112034-12115027	LRR Kinase	[111,139,142]
	*Glyma.02g122000*	*Glyma02g13470*	Gm02:12115284-12118493	LRR Kinase	[111,139,142]
	NM_001249088 ***	-	-	dsRNAse	[157]
	NM_001253944 ***	-	-	dsRNAse	[111]
	*Glyma.02g122100*	*Glyma02g13495*	Gm02:12134374-12137612	Cu Transport	[158]
	*Glyma.02g122200*	*Glyma02g13520*	Gm02:12141974-12149160	Chaperone DnaJ	[142]
	*Glyma.02g122300*	-	Gm02:12143960-12145950	Unknown	[142]
	*Glyma.02g122400*	*Glyma02g13530*	Gm02:12150906-12151220	Unknown	[142]
	*Glyma.02g122500*	*Glyma02g13540*	Gm02:12158735-12163084	tRNA Transferase	[142]
	*Glyma.02g122900*	*Glyma02g13590*	Gm02:12259463-12264960	BSD Domain	[111]
	*Glyma.02g123700*	*Glyma02g13700*	Gm02:12351993-12355050	Protein Kinase	[111]
	*Glyma.02g124300*	*Glyma02g13770*	Gm02:12425993-12427856	MYB Domain	[111]
	*Glyma.02g127800*	*Glyma02g14160*	Gm02:13010651-13015848	LRR Kinase	[159]
	*Glyma.02g128000*	*Glyma02g14190*	Gm02:13048160-13051248	Decarboxylase	[159]
	*Glyma.02g128200*	*Glyma02g14200*	Gm02:13093448-13095566	Methyltransferase	[159]

^†^ Gene correspondence name based on first and second sequencing assembly; -, no reported correspondence between genome assemblies; * *3gG2* gene name; ** *Rsvg2* gene name; *** open reading frames that do not exist in Wm82.a2.v1. ^‡^ Gene location based on the Williams 82 reference genome Wm82.a2.v1 assembly. ^§^ Predicted protein function based on domain.

**Table 4 viruses-14-01122-t004:** Summary molecular markers and candidate genes for resistance to *Soybean mosaic virus* strains in China.

Locus	SMV Strain	Marker	Marker/Interval Position (Wm82.a2)	Candidate Genes	Reference
*Rsc3*	SC3	-	-	*Glyma.13g190000*, *Glyma.13g190300*, *Glyma.13g190400*, *Glyma.13g190800*	[135]
*Rsc3Q*	SC3	BARCSOYSSR_13_1114 BARCSOYSSR_13_1136	Gm13:29815342-29815386 Gm13:30464888-30464941	*Glyma.13g187600*, *Glyma.13g187900*, *Glyma.13g190300*, *Glyma.13g190400*, *Glyma.13g190800*	[168]
*Rsc12*	SC12	Satt334 Sct_033	Gm13:29609521-29609720 Gm13:31355515-31355673	-	[169]
*Rsc14Q*	SC14	Sat_234 Satt334 Sct_033	Gm13:27656933-27657028 Gm13:29609521-29609720 Gm13:31355515-31355673	-	[169] [170]
*Rsc18*	SC18	SOYHSP176 Satt334 Bin65	- Gm13:29609521-29609720 Gm13:24850727-25266083	- *Glyma.13g150000*, *Glyma.13g151100*	[159] [171]
*Rsc20*	SC20	BARCSOYSSR_13_1099 BARCSOYSSR_13_1185	Gm13:29609605-29609634 Gm13:31348327-31348350	*Glyma.13g194700*, *Glyma.13g195100*	[172]
*Rsc4*	SC4	BARCSOYSSR_14_1413 BARCSOYSSR_14_1416	Gm14:46944330-46944387 Gm14:47007588-47007611	*Glyma.14g204600*, *Glyma.14g205000*, *Glyma.14g205200*	[139]
*Rsc5*	SC5	Bin352 Bin353	- -	Eleven candidate genes, including *Glyma.02g122100*	[158]
*Rsc7*	SC7	Satt634 Satt266 BARCSOYSSR_02_0621 BARCSOYSSR_02_0632 BARCSOYSSR_02_0667 BARCSOYSSR_02_0670	Gm02:11778567- 11778605 Gm02:14288260-14288310 Gm02:12156250-12156307 Gm02:12313357-12313394 - -	- Fifteen candidate genes	[162] [143] [173]
*Rsc8*	SC8	BARCSOYSSR_02_0610 BARCSOYSSR_02_0616 ZL-42 ZL-52	Gm02:11964524-11964549 Gm02:12070465-12070492 Gm02:11777821-11778102 Gm02:11778123-11808708	*Glyma.02g120700*, *Glyma.02g120800*, *Glyma.02g121500*, *Glyma.02g121900*, *Glyma.02g122000* *Glyma.02g121500*, *Glyma.02g121600*	[155] [174]
*Rsc13*	SC13	BARCSOYSSR_02_0610 BARCSOYSSR_02_0621	Gm02:11964524-11964549 Gm02:12156250-12156307		[173]
*Rsc18*	SC18	BARCSOYSSR_02_0667 BARCSOYSSR_02_0670	Gm02:13026964-13027011 Gm02:13104630-13104653	*Glyma.02g127800*, *Glyma.02g128200*, *Glyma.02g128300*	[159]
*Rsc15*	SC15	Sat_213 Satt286	Gm06:14644152-14644338 Gm06:16221044-16221254		[175]

**Table 5 viruses-14-01122-t005:** Williams and Essex near isogenic soybean lines carrying different SMV resistance genes/alleles.

Line ^†^	PI ^‡^	Locus	Pedigree
L78-379	PI 547844	*Rsv1*	Williams (6) × PI 96983
L83-542	-	*Rsv1*	F3 from BC5 Williams(6) × Buffalo
L83-551	-	*Rsv1*	F3 from BC5 Williams(6) × Buffalo
L96-1676	-	*Rsv1*	Williams (6) × Buffalo
L96-1680	-	*Rsv1*	Williams (6) × Buffalo
L96-1683	-	*Rsv1*	Williams (6) × Buffalo
L96-1687	-	*Rsv1*	Williams (6) × Buffalo
L88-8431	PI 547885	*Rsv1-r*	Williams (6) × Raiden
L88-8440	PI 547886	*Rsv1-r*	Williams (6) × Raiden
L92-8151	-	*Rsv1-s*	Williams (6) × PI 486355
L92-8580	PI 591516	*Rsv1-h*	Williams (6) × Suweon 97
L93-3327	PI 591515	*Rsv1-t*	Williams (6) × Ogden
L84-2112	PI 591513	*Rsv1-m*	Williams x ((Williams (6) × Marshall)
L85-2308	PI 547873	*Rsv1-y*	Williams (6) × Dorman
L86-1525	-	*Rsv3*	Williams (6) × Hardee
V94-3971	-	*Rsv1*	Essex (5) × PI 96983
V262	-	*Rsv1-n*	Essex (5) × PI 507389
V229	-	*Rsv3*	Essex (5) × L29
V94-5152	PI 596752	*Rsv4*	Essex × PI 486355
V97-9001	-	*Rsv4*	Essex (5) × PI 486355
V97-9003	-	*Rsv4*	Essex (5) × PI 486355

^†^ L-series of Williams isogenic lines developed by R.L. Bernard, Dep. of Crop Sciences, University of Illinois, 1101 West Peabody Dr., Urbana, IL 61810; V-series of Essex isogenic lines developed by G.R. Buss, Dep. of Crop and Soil Environmental Sciences, Virginia Polytechnic Institute and State University, Blacksburg, VA 24061. ^‡^ PI, plant introduction assigned for the isogenic line.

## Data Availability

Not applicable.

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
