# Peer review of "Decades of Genetic Research on Soybean mosaic virus Resistance in Soybean"

_viruses, 2022, doi:10.3390/v14061122_

Round 1

Reviewer 1 Report

Soybean mosaic virus (SMV) causes severe yield losses and seed quality reduction in soybean (Glycine max) production worldwide. This review summarizes histories and progress of research on characteristics of Soybean mosaic virus, host resistance and interactions between SMV and soybean. It is very helpful for soybean researchers to understand related knowledge. I think it can be accepted for publication after minor revise. List below are my comments which may be helpful for authors to improve the manuscript.

 Authors may take the information I provided here for their references.

1.The paper for resistance genes pyramiding has been published in 2017. Please refer to WANG Da-gang, et al. Marker-assisted pyramiding of soybean resistance genes RSC4, RSC8 and RSC14Q to soybean mosaic virus. Journal of Integrative Agriculture 2017, 16(0): 60345-7.

The main results of this paper are as follows.

Three SMV resistance genes, RSC4, RSC8, and RSC14Q, have been identified and mapped on soybean chromosomes 14, 2, and 13 from Dabaima, Kefeng 1, and Qihuang 1 cultivars, respectively. Soybean cultivar Nannong 1138-2 is widely grown in the Yangtze River valley of China. The crosses were made between (Qihuang 1×Kefeng 1) and (Dabaima×Nannong 1138-2). Ten simple sequence repeat (SSR) markers linked to three resistance loci (RSC4, RSC8, and RSC14Q) were used to assist pyramided breeding. Pyramided families containing three resistance loci (RSC4, RSC8, and RSC14Q) were evaluated by inoculating them with 21 SMV strains from China. Five F7 homozygous pyramided families exhibited resistance to 21 strains of SMV.

  1. Isuggest to pay attention to the paper of Yin, et al. A cell wall-localized NLR confers resistance to Soybean mosaic virus by recognizing viral-encoded cylindrical inclusion protein, Molecular Plant, https://doi.org/10.1016/j.molp.2021.07.13.

The main results of this paper are as follows.

  • Rsc4-3 in chromosome 14 is confirmed to confer resistance to SMV. It mediates resistance to multiple SMV strains such as SC3, SC7, SC8, SC11, SC14, G5 and G7.
  • Rsc4-3 encodes a cell-wall-localized NLR-type resistant protein
  • palmitoylation of Rsc4-3 is essential for the resistance.
  • cylindrical inclusion (CI) protein partially localizes to the cell wall and can interact with Rsc4-3.
  • CI is the avirulent gene for Rsc4-3-mediated resistance. demonstrating a gene-for-gene relationship.

Author Response

Comments and Suggestions for Authors

Soybean mosaic virus (SMV) causes severe yield losses and seed quality reduction in soybean (Glycine max) production worldwide. This review summarizes histories and progress of research on characteristics of Soybean mosaic virus, host resistance and interactions between SMV and soybean. It is very helpful for soybean researchers to understand related knowledge. I think it can be accepted for publication after minor revise. List below are my comments which may be helpful for authors to improve the manuscript.

Authors may take the information I provided here for their references.

1.The paper for resistance genes pyramiding has been published in 2017. Please refer to WANG Da-gang, et al. Marker-assisted pyramiding of soybean resistance genes RSC4, RSC8 and RSC14Q to soybean mosaic virus. Journal of Integrative Agriculture 2017, 16(0): 60345-7.

The main results of this paper are as follows.

Three SMV resistance genes, RSC4, RSC8, and RSC14Q, have been identified and mapped on soybean chromosomes 14, 2, and 13 from Dabaima, Kefeng 1, and Qihuang 1 cultivars, respectively. Soybean cultivar Nannong 1138-2 is widely grown in the Yangtze River valley of China. The crosses were made between (Qihuang 1×Kefeng 1) and (Dabaima×Nannong 1138-2). Ten simple sequence repeat (SSR) markers linked to three resistance loci (RSC4, RSC8, and RSC14Q) were used to assist pyramided breeding. Pyramided families containing three resistance loci (RSC4, RSC8, and RSC14Q) were evaluated by inoculating them with 21 SMV strains from China. Five F7 homozygous pyramided families exhibited resistance to 21 strains of SMV.

  1. I suggest to pay attention to the paper of Yin, et al. A cell wall-localized NLR confers resistance to Soybean mosaic virus by recognizing viral-encoded cylindrical inclusion protein, Molecular Plant, https://doi.org/10.1016/j.molp.2021.07.13.

The main results of this paper are as follows.

  • Rsc4-3 in chromosome 14 is confirmed to confer resistance to SMV. It mediates resistance to multiple SMV strains such as SC3, SC7, SC8, SC11, SC14, G5 and G7.
  • Rsc4-3 encodes a cell-wall-localized NLR-type resistant protein
  • palmitoylation of Rsc4-3 is essential for the resistance.
  • cylindrical inclusion (CI) protein partially localizes to the cell wall and can interact with Rsc4-3.
  • CI is the avirulent gene for Rsc4-3-mediated resistance. demonstrating a gene-for-gene relationship.

Authors comments:

Thank you for your time and effort to review this manuscript. We appreciate the addition of two new references suggested. We added short summaries to this manuscript.

Reviewer 2 Report

  1. I will suggest authors change the last two keywords as they are very general ones, including keywords related to SMV.
  2. I will suggest authors include a paragraph for the reason to construct the present review in the initial part.
  3. I will suggest Authors include the last paragraph under the head Conclusion.
  4. As I can see, a significant part of the present review focused on Rsv I will suggest authors include it in the title of the review.

Author Response

Comments and Suggestions for Authors

  • I will suggest authors change the last two keywords as they are very general ones, including keywords related to SMV.
  • I will suggest authors include a paragraph for the reason to construct the present review in the initial part.
  • I will suggest Authors include the last paragraph under the head Conclusion.
  • As I can see, a significant part of the present review focused on Rsv I will suggest authors include it in the title of the review.

Authors comments:

Thank you for your time and effort to review this manuscript. The keywords were changed to “virus resistance”, “soybean breeding”, “genetic diversity”. The paragraph has been added as suggested. Also, in the paragraph about the reason to construct the present review, we added a note that this review is focused mostly on the Rsv genes but not limited to the resistance to U.S.A. strains.